# Research on Bearing Surface Scratch Detection Based on Improved YOLOV5

**DOI:** 10.3390/s24103002

**Published:** 2024-05-09

**Authors:** Huakun Jia, Huimin Zhou, Zhehao Chen, Rongke Gao, Yang Lu, Liandong Yu

**Affiliations:** College of Control Science and Engineering, China University of Petroleum (East China), Qingdao 266580, China

**Keywords:** bearing surface scratches, improved YOLOV5, defect detection

## Abstract

Bearings are crucial components of machinery and equipment, and it is essential to inspect them thoroughly to ensure a high pass rate. Currently, bearing scratch detection is primarily carried out manually, which cannot meet industrial demands. This study presents research on the detection of bearing surface scratches. An improved YOLOV5 network, named YOLOV5-CDG, is proposed for detecting bearing surface defects using scratch images as targets. The YOLOV5-CDG model is based on the YOLOV5 network model with the addition of a Coordinate Attention (CA) mechanism module, fusion of Deformable Convolutional Networks (DCNs), and a combination with the GhostNet lightweight network. To achieve bearing surface scratch detection, a machine vision-based bearing surface scratch sensor system is established, and a self-made bearing surface scratch dataset is produced as the basis. The scratch detection final Average Precision (AP) value is 97%, which is 3.4% higher than that of YOLOV5. Additionally, the model has an accuracy of 99.46% for detecting defective and qualified products. The average detection time per image is 263.4 ms on the CPU device and 12.2 ms on the GPU device, demonstrating excellent performance in terms of both speed and accuracy. Furthermore, this study analyzes and compares the detection results of various models, demonstrating that the proposed method satisfies the requirements for detecting scratches on bearing surfaces in industrial settings.

## 1. Introduction

Bearings are very common and critical components, as the core components of mechanical equipment, mainly used to support the mechanical rotating body and reduce the coefficient of friction. The efficiency and service life of mechanical equipment are significantly impacted by the quality and performance of bearings, so it is necessary to strengthen the quality inspection during the production and manufacturing process of bearings to ensure that the bearing qualification rate meets the requirements. Therefore, detecting scratches on the surface of the bearing [1] is crucial. The automated detection of bearing scratches has high research value.

Traditional methods for detecting scratches rely on manual feature extraction. However, due to the various shapes and directions of scratches, the accuracy of traditional methods is low, making it difficult to meet the requirements of on-site industrial inspection. In recent years, machine vision technology has been increasingly applied in industrial production and manufacturing. It completes the detection or identification of the products being measured, improving the production efficiency of the enterprise and reducing production costs. The application of machine vision technology in the field of defect detection is becoming increasingly prevalent. Currently, there are two principal categories of defect detection methods based on machine vision technology [2].

One approach uses traditional image processing algorithms, such as a histogram of the oriented gradient [3] (HOG) and deformable parts model [4,5,6] (DPM), typically involving three steps: region selection, feature extraction, and classification regression. However, manually extracting target features has limitations for detecting small targets such as scratches.

The other uses deep learning techniques. With the development of deep learning technology, target detection algorithms based on convolutional neural networks are generally formed into two-stage and single-stage detection algorithms.

Region convolutional neural networks [7] (RCNNs), as the starting point of target detection in the field of deep learning, are classical two-stage target detection algorithms, with a very important status and reference value. RCNN utilizes region proposals with the SS algorithm [8], feature extraction, object classification, bounding box refinement, and proposal filtering for target detection. Fast RCNN [9] and Faster RCNN [10] were proposed on this basis. Two-stage target detection also leads to high computational complexity and slow processing speed in two-stage networks, making them unsuitable for real-time target detection tasks.

Single-stage target detection networks use convolutional neural networks [11] (CNNs) to extract image features and directly predict the location and class of the target, which is fast but slightly less accurate, but following several generations of networks, the accuracy has been greatly improved. The YOLO [12,13,14,15,16,17,18,19] family is the most representative network for single-stage object detection. You Only Look Once (YOLOV1) performs object detection by dividing the image into a grid and predicting bounding boxes and class probabilities directly from grid cells. Researchers have continued to improve the YOLO family of target detection networks and have developed a number of versions.

Research on bearing defect detection has also evolved from using traditional image algorithms to deep learning methods. Zhengyan Gu et al. [20] proposed a machine vision-based automatic detection and identification method for bearing surface defects. Tise method improves and combines the Ostu algorithm and the Canny algorithm to enhance the completeness and accuracy of bearing surface defect segmentation. Dan L et al. [21] proposed a deep learning method for bearing face detection based on data enhancement and improved fast RCNN, using a semi-supervised data enhancement approach based on local flaw features, the improved RA strategy, and the mosaic algorithm to enhance the initial bearing sample data. This method can effectively achieve accurate and fast bearing face flaw detection. Dahai L et al. [22] proposed a non-destructive testing (NDT) method for detecting surface defects on cylindrical rollers of silicon nitride bearings. The method is based on an optimized convolutional neural network that combines a semantic segmentation sub-network and a decision-making sub-network. It achieves high levels of speed and accuracy. Liu B et al. [23] designed a machine vision system that uses a multi-angle light source to capture clear images of surface defects on bearings. They achieved the automatic detection of small defects through contour sub-area fitting and the improved Nobuyuki OTSU method. Zheng Z et al. [24] employed a deep learning approach to detect surface defects on bearing caps. They improved the YOLOV3 network structure by proposing the BNA-Net feature extraction network, attention prediction sub-network, and defect localization sub-network. The final AP value for defect detection reached 69.74%. It can be seen that current research on bearing defect detection, despite being combined with deep learning technology, has not achieved high detection accuracy and requires improvements in efficiency to meet the industrial demand.

This study focuses on bearing surface scratch detection. A scratch detection model named YOLOV5-CDG is proposed, which is based on the improved YOLOV5 to achieve efficient and high-accuracy detection of scratches on bearing surfaces, and a homemade dataset is used to complete experimental validation. Our contributions are summarized as follows:
(1)We designed a machine vision-based bearing surface scratch sensor system that meets the needs of bearing surface scratch inspection. By utilizing the system, a novel dataset of bearing surface scratches was produced, providing high-quality bearing surface scratch images to support our experiments.(2)Based on the YOLOV5 model, we proposed a novel model that improves the performance of YOLOV5 by adding the Coordinate Attention (CA) mechanism module, incorporating Deformable Convolutional Network (DCN) feature extraction and employing the GhostNet network, and we named it YOLOV5-CDG. The model achieved efficient and high-accuracy detection of scratches on bearing surfaces.

This paper is organized as follows: Section 2 introduces the modules used in our proposed YOLOV5-CDG model. Section 3 describes the sensor system we established and the bearing surface scratch dataset we produced, launching a series of experiments to test the performance of the YOLOV5-CDG model. Section 4 analyzes the experimental results and verifies the excellent performance of the YOLOV5-CDG model. Finally, Section 5 discusses this study.

## 2. Materials and Methods

Section 2.1 provides a concise overview of the YOLOV5 model, based on which we incorporate the CA, DCN, and GhostNet modules and present them in Section 2.2, Section 2.3, and Section 2.4, respectively. The YOLOV5-CDG model is then proposed in Section 2.5.

### 2.1. Overview of YOLOV5

YOLOV5, compared to the original version, has a significant improvement in the accuracy of target detection and speed compared with the original version, and it can be easily deployed into embedded or CPU devices. Therefore, in this study, we ultimately carry out bearing surface scratch detection based on YOLOV5 target detection network.

The YOLOV5 network architecture is composed of four main parts: input, backbone, neck, and prediction. The network divides the image into grids and predicts the presence of a target, along with its category and location information, for each grid. YOLOV5 performs a single forward computation to obtain the target detection results, making it faster than two-stage target detection networks. This study introduces and improves upon the YOLOV5 model, specifically YOLOV5s-v6.1. All subsequent references to YOLOV5 refer to YOLOV5s-v6.1. Figure 1 illustrates the network structure of YOLOV5, with the SPPF module highlighted in a dashed rectangular box. The convolution module is labelled with kernel size, stride, and padding.

The YOLOV5 backbone consists of CSPDarknet53, which was initially proposed in YOLOV3. The network comprises 53 convolutional layers, which refer to the convolutional layer, the BN layer, and the SiLU activation function. The BN layer normalizes the data to alleviate the problems of gradient explosion and gradient vanishing and speeds up the network’s convergence. The SiLU activation function is a weighted linear combination of Sigmoid activation functions. Figure 2 displays the structure of the C3 module in YOLOV5. The first convolution module employs a kernel size of 6 × 6. The C3 module comprises three convolution modules and a BottleNeck module. CSPDarknet53 introduces the CSP [25] structure, which is primarily based on Darknet53. The CSP structure, compared to the traditional convolution structure, divides the shallow feature map into two parts in the channel dimension. The first part is followed by direct splicing with the resulting second part without any additional processing. This approach significantly reduces the reuse of gradient information, resulting in a 10% to 20% reduction in network parameters while maintaining network accuracy. In the backbone, the BottleNeck part of the C3 module uses the BottleNeck1 structure, while the remaining part of the C3 module uses the BottleNeck2 structure.

The structure of the neck part is based on the SPPF module and CSP-PAN [26], with improvements made to the SPPF module. The SPP module implements spatial pyramid pooling by adopting the idea of SPPNet [27]. This involves passing the input feature maps through pooling layers with different kernel sizes in parallel, achieving multi-scale feature fusion to some extent. The SPPF passes the input feature layer sequentially through the pooling layer with a kernel size of 5 × 5 and concatenates the output of each pooling layer. However, multiple serial pooling operations with the same kernel size are equivalent to pooling with a larger kernel size, resulting in the same output and improved network efficiency. The YOLOV5 PAN is based on the FPN [28] network. The Feature Pyramid Network (FPN) fuses low-level and high-level features, providing rich semantic information at all scales and effectively addressing the multi-scale problem of image features. FPN integrates semantic features into the low-level feature map but does not consider location information. PAN adds a bottom-up pyramid structure based on FPN, which transfers strong localization information from low to high levels. YOLOV5 also uses the CSP structure in PAN, which improves localization accuracy when using high-level features for target recognition.

The Prediction section includes three detection heads that operate on different predicted feature maps. The detection heads are responsible for predicting large, medium, and small targets based on the size of the predicted feature maps. Using the input image size of 640 × 640 as an example, the feature maps of the three prediction heads are 80 × 80, 40 × 40, and 20 × 20, respectively. These maps are used to predict small, medium, and large targets, in that order. Each prediction sign map is followed by outputting the prediction parameters using 1 × 1 convolution. The prediction parameters include target category, anchor box position, and confidence score. For each target category, the confidence score indicates the probability that the anchor box contains a target of that category. Finally, the anchor box with confidence scores that exceed the threshold value is output in order to complete the detection of the target object.

### 2.2. Coordinate Attention 

The attention mechanism (AM) [29] is a technique that deep learning models use to mimic the way humans allocate their attention when processing information. This technique enables the model to focus on the most important information for the current task. In target detection, the attention mechanism can assist the model in identifying and locating the target more accurately. This allows the model to focus more on regions that may contain targets during target detection, thereby enhancing the accuracy and efficiency of detection.

Attention mechanisms can be classified into spatial domain, channel domain, and hybrid domain attention mechanisms. SE [30] attention mechanism only considers the importance between channels and does not consider the spatial coordinate information. The Coordinate Attention (CA) [31] mechanism, proposed by Hou Q et al. in 2021, combines channel attention with location information to enhance the performance of the network model. It generates attention weights and assigns smaller weights to non-target regions, embedding location information within the channels. As shown in Figure 3, the CA attention mechanism introduces two one-dimensional feature encoders to extract the perceptual attention feature maps in the horizontal and vertical directions, respectively. Set the input features as C × H × W. C, H, and W represent the number of channels, the height, and the width of the input feature map, respectively. Separating the processing of spatial and channel information in advance can effectively preserve spatial information while taking into account channel information for subsequent combination. The input features are first average pooled in the horizontal and vertical directions, resulting in feature maps of dimensions C × H × 1 and C × 1 × W. These two feature maps are then concatenated. The feature maps in the horizontal and vertical directions are obtained through the operations of slicing and normalization. Finally, the obtained feature maps in the horizontal and vertical directions are fused with the original input feature maps to generate the feature maps with attention weights.

The use of the CA attention mechanism can improve the model’s ability to locate and recognize targets, particularly in tasks involving spatial relationships. This technique reduces the impact of unnecessary information and noise, improving the model’s robustness and generalization performance. Additionally, it enhances the accuracy of the model. In target detection tasks, the CA attention mechanism improves the ability to locate the target and increases detection accuracy. Finally, the CA attention mechanism improves computational efficiency. It does not significantly increase the number of parameters in the network and can improve model performance. This makes the CA attention mechanism more feasible for real-world applications. In conclusion, this technique has the potential to enhance model performance and improve a variety of visual tasks.

### 2.3. Deformable Convolutional Networks

Traditional convolutional operation samples input features within the receptive field in a fixed manner, which may not adapt well to the deformation of the target object. Deformable Convolutional Network (DCN) [32], proposed by Dai, J et al. in 2017, is a module of convolutional neural networks for computer vision tasks; it performs the convolutional operation with the added capability of local deformation by introducing a deformable convolutional kernel. This kernel learns a set of offsets, allowing it to adaptively sample input features based on their local context, resulting in better adaptation to the deformation of the target object. Therefore, this study proposes the fusion of deformable convolution in the YOLOV5 feature extraction module. As shown in Figure 4, the blue dots represent the 3 × 3 convolution kernel sampling positions, and the green dots represent the sampling positions after the convolution kernel offset.

Using a 3 × 3 convolution kernel as an example, the convolution computation can be represented by the set *R*: (1)R={(−1,−1),(−1,0),…,(0,1),(1,1)}

Assuming *x* as the input and *y* as the output, the convolution operation for the current pixel point *p*_0_ can be expressed as:(2)$$y\left(p_{0}\right)=\underset{p_{n}\in R}{\sum }w\left(p_{n}\right)\cdot x\left(p_{0}+p_{n}\right)$$

In deformable convolution, an additional offset ∆*p_n_* can be added to Equation (9), where *p_n_* is the offset of *p*_0_ relative to the receptive field within the set *R*, *p_n_* is an integer, and *w* is the sampling weight.
(3)$$y\left(p_{0}\right)=\sum_{p_{n}\in R}w\left(p_{n}\right)\cdot x\left(p_{0}+p_{n}+\Delta p_{n}\right)$$

When calculating ∆*p_n_* from another convolution, it is typically not an integer. As a result, *x*(*p*_0_ + *p_n_* + ∆*p_n_*) does not correspond to an actual integer pixel point in the image. Therefore, bilinear interpolation is used to calculate it.
(4)$$p=p_{0}+p_{n}+\Delta p_{n}$$
(5)$$x(p)=\underset{q}{\sum }G(q,p)\cdot x(q)=\underset{q}{\sum }g\left(q_{x},p_{x}\right)\cdot g\left(q_{y},p_{y}\right)\cdot x(q)$$
(6)g(a,b)=max(0,1−|a−b|)

The formula for bilinear interpolation is represented by *G(.*,*.)*, where *x*(*q*) is the value of the entire pixel point on the feature map. The weight coefficient *g*(*a*, *b*) is computed based on the coordinates and denotes the distance between the two coordinate points, with a maximum value of 1. The physical significance of this is that the coordinate point q within one pixel from the horizontal and vertical coordinates of the p-points participates in the operation, resulting in the value of the p-point.

The process described above does not take into account the size of the offset. When the offset is too large, the convolution kernel may deviate from the target region. To address this issue, DCNv2 [33] adds a modulation module to control the degree of offset change. The original Equation (10) is modified to include weight coefficients based on the offset. The specific formula is as follows:(7)$$y(p_{0})=\sum_{p_{n}\in R}w\left(p_{n}\right)\cdot x\left(p_{0}+p_{n}+\Delta p_{n}\right)\cdot \Delta m_{n}$$

The value of ∆*m_n_* is set within a range of 0 to 1, requiring a separate convolution to learn the parameters.

Figure 5 illustrates the deformable convolution process, where an offset region is generated after the convolution calculation, and the offset is applied to the original convolution kernel, i.e., deformable convolution.

The scratches on bearings are typically elongated. To extract scratch features more efficiently, deformable convolution can be added in the feature extraction stage. This avoids the extraction of redundant information by the traditional fixed convolution kernel. Additionally, it improves target detection accuracy. The DCN module captures deformation information and precise positional alignment of the target, improving target location and identification while reducing missed and false detections.

### 2.4. GhostNet

Traditional deep learning models employ a large number of convolutional parameters, resulting in long inference times. This makes it challenging to deploy them in industrial settings for real-time detection. To address this issue, a potential solution is to create lightweight deep convolutional models. The GhostNet [34] is a new lightweight network that reduces the number of parameters by generating redundant feature maps through a concise computational approach. This is achieved by introducing the Ghost module, which consists of two sub-networks: MainNet and GhostNet. The MainNet extracts the main feature representation, while the GhostNet performs lighter-weight computations to extract additional features and fuse them with those of the MainNet.

Figure 6 illustrates that the Ghost module comprises two operation parts. The first step involves using ordinary convolution to generate a small number of channels of the real feature map from the input image or feature map. The second step involves performing simple linear operations on the feature map obtained in the first step to obtain the Ghost feature layer. The feature layer is combined with the Ghost feature layer, resulting in a final output feature map with the same number of channels as a standard convolution operation.

The size of the input feature map is represented by *h* × *w* × *c*, where *h* is the height, *w* is the width, and *c* is the number of channels. The size of the output feature map is denoted by *h′* × *w′* × *c′*, and the size of the conventional convolution kernel is *k* × *k*. The operation of the kernel is as follows:
(8)$$cost_{1}=h^{\prime }\times w^{\prime }\times n\times k\times k\times c$$
where *c* is the number of input channels, *h* is the height of the input map, and *w* is the width of the input map.

The output feature map channel *n* is divided into *s* equal parts, with the number of ordinary convolution output channels accounting for 1/*s*. The Ghost module’s linear operation is considered as deep convolution, with the deep convolution kernel size being *d × d* and the number of output channels accounting for (1 − *s*)/*s*.
(9)$$cost_{2}=h^{\prime }\times w^{\prime }\times \frac{n}{s}\times k\times k\times c+(s-1)\times h^{\prime }\times w^{\prime }\times \frac{n}{s}\times d\times d$$

The speed ratio of the normal convolution to the Ghost module is:(10)$$\begin{array}{ll} r_{s} & =\frac{n\cdot h^{\prime }\cdot w^{\prime }\cdot c\cdot k\cdot k}{\frac{n}{s}\cdot h^{\prime }\cdot w^{\prime }\cdot c\cdot k\cdot k+(s-1)\cdot \frac{n}{s}\cdot h^{\prime }\cdot w^{\prime }\cdot d\cdot d}\\ & =\frac{c\cdot k\cdot k}{\frac{1}{s}\cdot c\cdot k\cdot k+\frac{s-1}{s}\cdot d\cdot d}\approx \frac{s\cdot c}{s+c-1}\approx s, \end{array}$$

When both *d* × *d* and *k* × *k* are convolutional kernel sizes, their values can be considered approximately equal. Additionally, since *s* is much smaller than *c*, the ratio of operations can be approximated to be *s*. The analysis above shows that the Ghost module can significantly reduce the arithmetic and parametric quantities of a model compared to the traditional convolutional module. It has also demonstrated comparable or even superior performance to larger and more complex models in various experimental species for multiple image classification and target detection tasks. Thus, this study employs the Ghost module to substitute certain conventional convolutional modules, thereby enhancing the model’s overall performance.

### 2.5. YOLOV5-CDG

The CA attention mechanism is added to the last layer of the YOLOV5 feature extraction layer to enable it to pay more attention to the target region during network training. This results in less weight being given to irrelevant regions. The original YOLOV5 network model is improved through this addition. The deformable convolution replaces some of the convolutions in the feature extraction network by changing the sampling position of the convolution kernel to make it closer to the shape of the scratch. The Ghost module reduces the number of parameters in the network by generating Ghost layers. Therefore, the convolution module of the C3 module in YOLOV5 is replaced with the Ghost module. The network structure is shown in Figure 7, and we name the improved YOLOV5 network YOLOV5-CDG; the highlighted section of the border represents the enhancement of YOLOV5-CDG.

## 3. Experiments

Section 3.1 describes the selection and construction of the hardware for the construction of the sensor system, Section 3.2 outlines the self-made experimental dataset, while Section 3.3 details the experimental methodology employed in this study.

### 3.1. Machine Vision-Based Bearing Surface Scratch Sensor System

We examined the core hardware used to form the sensor system, which includes industrial cameras, lenses, and light sources. To ensure the accuracy of the system, it is essential to analyze the hardware selection. In this section, we examine the three main types of hardware and consider a range of factors, including cost, function, and other aspects, to determine the hardware models that are suitable for research in this study and the bearing surface defect detection sensor system we established with them.
(1)Industrial camera selection: In consideration of the task requirements set forth in this study, which pertain to the shooting of bearings, it was determined that the size of the bearing, working distance, and the selection of suitable cameras would be of paramount importance. We ultimately selected the industrial camera MV-CA050-12UC (Hikvision, Hangzhou, China), a 5-megapixel surface array camera. The detailed parameters of the camera are presented in Table 1. (2)Lens selection: In consideration of the subject, the camera, and other pertinent factors, the OPTO double telecentric lens TC23036 was selected. The lens’s detailed parameters are presented in Table 2. (3)Light source selection: We opted for the OPT-CO80-B coaxial light source. The specific parameters of this light source are presented in Table 3. The coaxial light source is a blue coaxial light source with a luminous surface area of 73 × 73 mm. By positioning the coaxial light source on top of the bearing and capturing an image of the bearing surface, the scratches on the bearing surface can be highlighted.

Based on these hardware systems and Python (Versions: 3.8.16) language programming to implement the sensor system bearing surface defect detection function, the sensor system built in the laboratory is shown in Figure 8, where the bearing is placed on the carrier table with a coaxial light source above the bearing. MVS3.4.1 (Hikvision, Hangzhou, China) machine vision industrial camera client is installed on the computer to connect and communicate between the computer and the camera. When the image of the bearing surface is collected, the coaxial light source is turned on.

### 3.2. Experimental Dataset

This section’s dataset comprises data from both company and homemade sources. The dataset of 1809 images of bearing surface scratches was collected using a camera, including both qualified and defective bearings. Figure 9 shows the bearing images before and after annotation.

The dataset is randomly divided into training, validation, and test sets, where the training set includes 1206 images, the validation set 206 images, and the test set 397 images, and the basic information of the dataset is shown in Table 4.

### 3.3. Experimental Method

Table 5 shows the experimental parameter configuration information.

To expand the training set when training the model, data enhancement methods are used due to the small dataset in this experiment. During the training process, one or more of the image enhancement methods mentioned above will be randomly selected to process the training data. Figure 10 demonstrates the effect of two images after random scaling, random panning, and random horizontal flipping.

The training process utilizes the Warmup training strategy. This strategy involves using a small learning rate at the beginning of training, gradually increasing it to a set learning rate, and then decreasing it again as the number of training rounds increases. The use of Warmup can solve the instability of training caused by an initial learning rate that is too large. Figure 11 illustrates the changes in learning rate during the training process. During training, the image is uniformly scaled to a size of 640 × 640. Pre-training weights YOLOV5s.pt are loaded to help the network converge faster. An epoch refers to one complete training set of the model using all the data from the training set. The experiments in this section are set up to train for 150 epochs. 

Defect detection in industry involves two steps: firstly, identifying defective products, and, secondly, labelling the location of the defect. The scratch detection experiments mentioned above were conducted solely on the scratch dataset. This is because placing a targetless image during the training of the target detection network hinders the network’s ability to efficiently extract target features. Therefore, all training and test data used bearing images with scratches, which allowed the network to extract scratch features more efficiently. Finally, the trained weights were used to test the classification accuracy of the network for both scratch-free qualifying bearings and scratch-defective bearings.

## 4. Experimental Results and Analysis

Section 4.1 analyzes the results of the ablation experiments on the detection performance and operational efficiency of the YOLOV5-CDG model in the dataset with scratches. Section 4.2 compares the performance of the different models. In Section 4.3, the model’s ability to detect bearings with and without passes on the full dataset is validated.

### 4.1. Analysis of Experimental Results of the YOLOV5-CDG Model

#### 4.1.1. Accuracy Results Analysis 

Experiments were conducted on the original YOLOV5 model and the improved YOLOV5 model separately to train the scratch detection model based on the prepared experimental data and environment. The CA attention mechanism, deformable convolution, and Ghost modules are added separately to the YOLOV5 network, along with the YOLOV5-CDG network proposed in this study, and trained and tested separately on the test set. The precision, recall, F1 score, and AP value are recorded for each experiment. Table 6 presents a comparison of the experimental results of each model.

Table 6 shows that training and testing on the bearing scratch dataset using the original YOLOV5 network resulted in a precision of 95.9%, a recall of 88%, an F1 score of 0.918, and an AP value of 93.6%. After adding the CA attention mechanism in front of the SPPF module of the original YOLOV5 network and fusing the DCN module in some of the traditional convolutions of the feature extraction module, all of the indexes were improved to a certain extent. It can be seen that the introduction of the CA module can effectively improve the network’s performance, which also proves the effectiveness of fusing deformable convolutions in the YOLOV5 network. By replacing all convolutional layers in the C3 module of the network with GhostNet, the network’s indexes decreased, albeit less than YOLOV5. After implementing all improvements, the YOLOV5-CDG model achieved an accuracy of 97.2% on the test set, a 1.3% improvement; a recall of 92.5%, a 4.5% improvement; an F1 score of 0.948, a 0.03 improvement; and an AP value of 97%, a 3.4% improvement. The experiments demonstrated that the YOLOV5-CDG model exhibits superior performance in detecting scratches on bearing surfaces.

#### 4.1.2. Speed Results Analysis

In addition to ensuring the accuracy of industrial inspection applications, it is also important to consider the speed of model calculation. As shown in Table 7, we recorded the number of parameters, the amount of computation, the CPU processing time, and the GPU processing time for each model.

After analyzing Table 7, it can be concluded that when training and testing the bearing scratches dataset using the original YOLOV5 network, the average inference time per image is 371.8 ms on the CPU device and 16.8 ms on the GPU device. The addition of the CA attention mechanism prior to the SPPF module improves the accuracy of the network without a significant increase in the number of parameters. The integration of deformable convolution into YOLOV5’s convolution allows the model to decrease picture inference time without sacrificing accuracy, resulting in an overall improvement in performance. Replacing the convolutional layer in the C3 module of the YOLOV5 backbone network with GhostNet leads to a significant reduction in the number of network parameters and computation, demonstrating the effectiveness of GhostNet in improving network speed. Overall, YOLOV5-CDG demonstrates superior performance in terms of inference speed, with a 29% reduction in inference time on the CPU device and a 27% reduction in inference time on the GPU device.

### 4.2. Performance Comparison of Different Models

As there are not many studies on bearing scratch detection and most of them, like ours, are based on their own established datasets, we briefly compare the accuracy of the different models, as shown in Table 8. The datasets are not exactly the same, but the sample sizes are sufficient, and the scratch patterns are similar, so the experimental result is representative to a certain extent.

### 4.3. Analysis of Bearing Qualification Test Results

Figure 12 shows the variation curve of AP for both the original YOLOV5 network and the method proposed in this study. Both networks were trained using the official YOLOV5 trained model, YOLOV5s.pt, as pre-training weights. It is evident that after continuous training, this study’s method outperforms YOLOV5 in terms of AP value and improves the accuracy of scratch detection.

Figure 13 shows the test bearing and the trained YOLOV5-CDG network weights file used to predict the location of bearing scratches. Three images are selected for demonstration, and the confidence levels of the (a), (b), and (c) plots are 0.88, 0.9, and 0.75, respectively. The scratches in all three images are accurately labelled. However, the scratches on test bearing 3 are not obvious, resulting in a low confidence level for the network.

The dataset for bearing testing comprises 256 images of bearings without scratches and 300 images of bearings with scratches. Of the images with scratches, 253 were correctly identified, while the remaining 3 were identified as not having scratches. All images without scratches were correctly identified. The dataset contains a total of 556 pictures, of which 553 were correctly identified, resulting in a classification accuracy of 99.46%.

The experiment demonstrates that the YOLOV5-CDG network achieves high classification accuracy for both qualified and defective bearings. The network’s AP value for detecting surface scratches on defective bearings is up to 97%, indicating that it can meet the needs of industrial field inspection in terms of detection speed and accuracy.

## 5. Discussion

An improved YOLOV5 bearing surface scratch defect detection network named YOLOV5-CDG is proposed in this study, achieving the detection of bearing surface scratches. The CA attention mechanism is incorporated into the feature extraction network, and some of the convolutional layers in the feature extraction network are fused into deformable convolutions. As the number of parameters increases, the accuracy of the network improves. The traditional convolutional module of the C3 module in the network is replaced with the Ghost module, resulting in a reduction in the number of parameters and computations of the network and a significant improvement in the inference speed. A homemade dataset is utilized to train the network, and its performance is evaluated using multiple metrics. The experimental results indicate an AP value of 97% for scratch detection, an accuracy of 99.46% for defective and qualified product detection, an average detection time of 263.4 ms per image on CPU devices, and 12.2 ms per image on GPU devices. Furthermore, a comparative analysis is conducted to compare the detection results of different models. The results validate that the proposed method, as compared to the original YOLOV5 network, achieves enhanced speed and accuracy, effectively meeting the requirements of bearing surface scratch detection in industrial sites.

Currently, there are limited studies and industrial datasets available for bearing scratch detection. It is important to consider the emerging noise and other actual industrial production situations. In the future, we plan to conduct research in more practical environments to optimize the algorithms and improve the model’s detection performance.

## Figures and Tables

**Figure 1 sensors-24-03002-f001:**
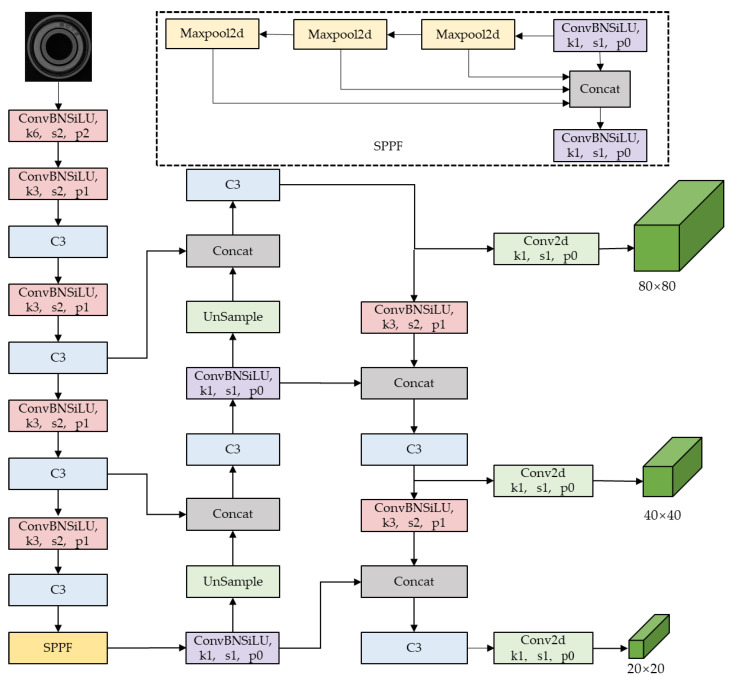
The network structure of YOLOV5.

**Figure 2 sensors-24-03002-f002:**
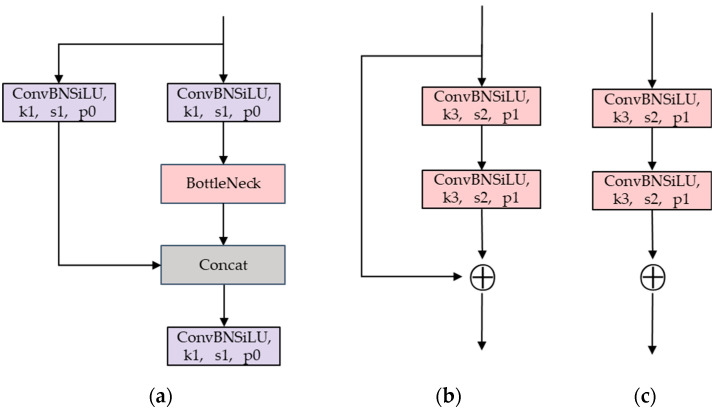
The structure of C3 module. (**a**) C3. (**b**) BottleNeck1. (**c**) BottleNeck2.

**Figure 3 sensors-24-03002-f003:**
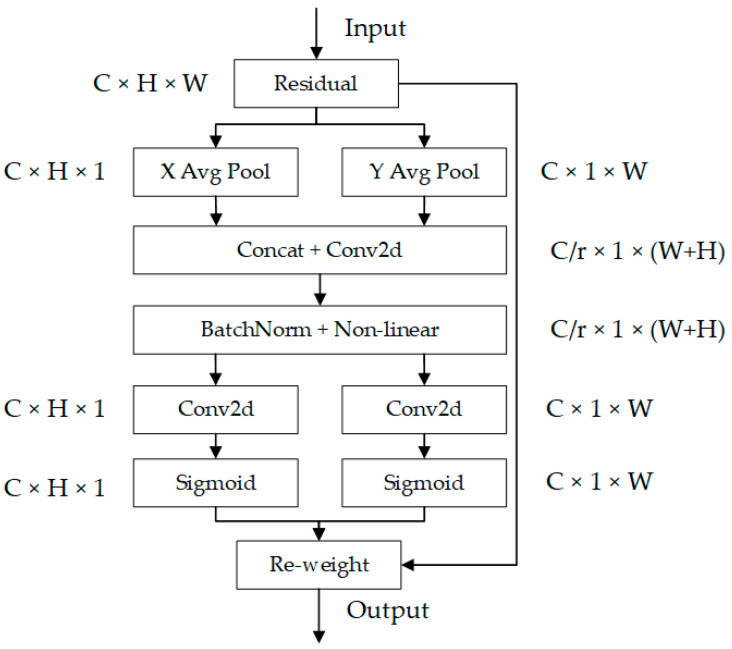
Coordinate Attention.

**Figure 4 sensors-24-03002-f004:**
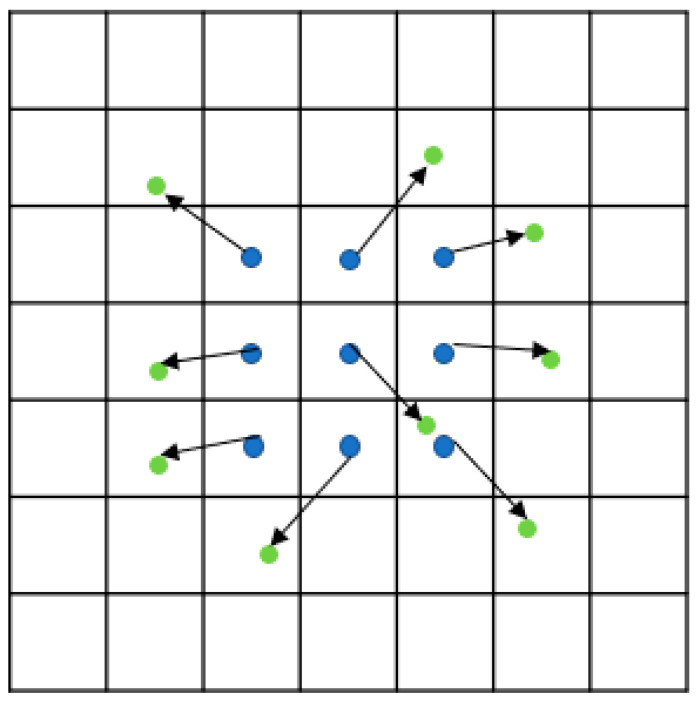
Illustration of the sampling locations in 3 × 3 convolutions.

**Figure 5 sensors-24-03002-f005:**
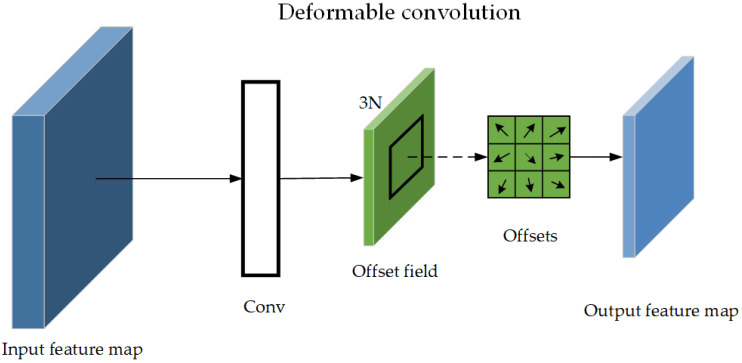
Structural diagram of deformable convolution.

**Figure 6 sensors-24-03002-f006:**
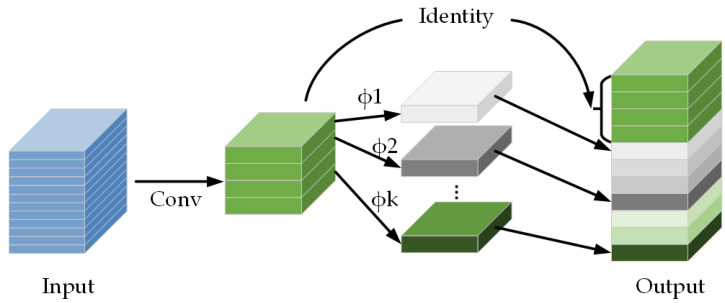
The Ghost module.

**Figure 7 sensors-24-03002-f007:**
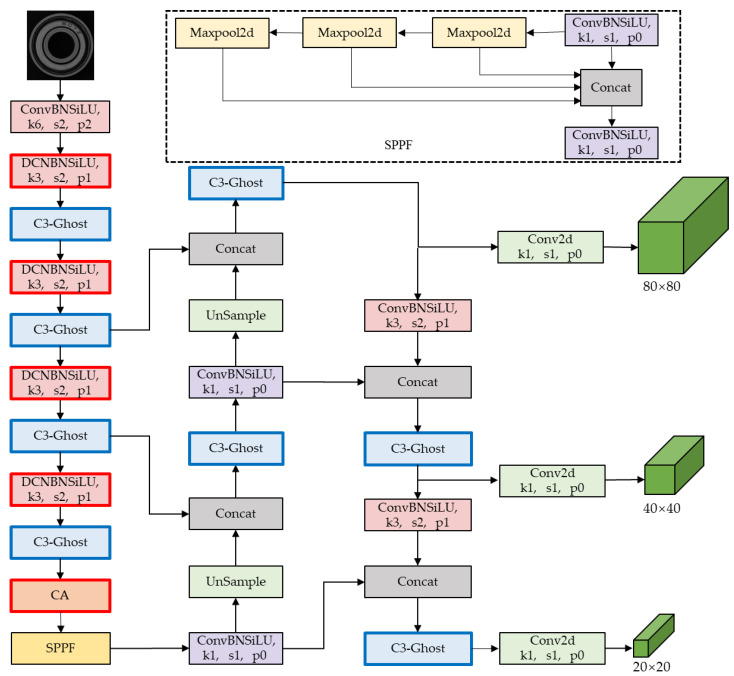
The YOLOV5-CDG network structure diagram of improved YOLOV5.

**Figure 8 sensors-24-03002-f008:**
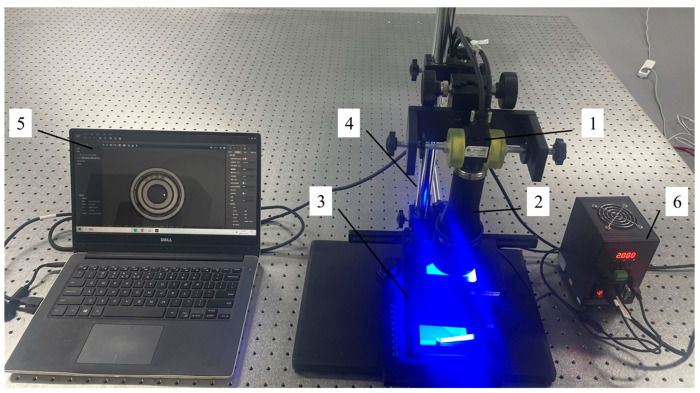
Bearing surface defect detection sensor system. (1) Industrial camera. (2) Lens. (3) Coaxial light source. (4) Experimental stand. (5) Host computer. (6) Light source controller.

**Figure 9 sensors-24-03002-f009:**
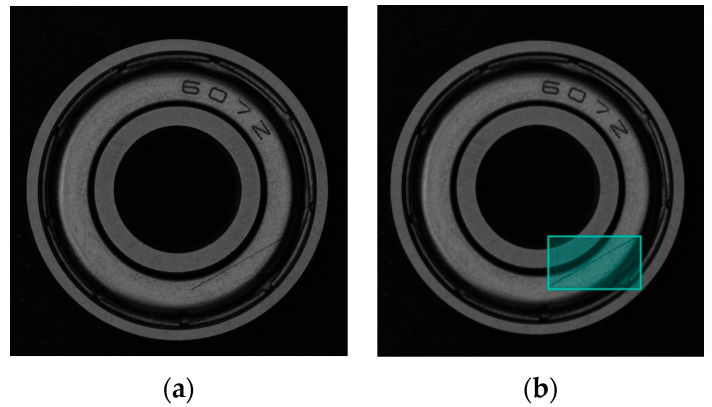
Marking of scratches on bearings. (**a**) Original bearing diagram. (**b**) Visualization of labelled information.

**Figure 10 sensors-24-03002-f010:**
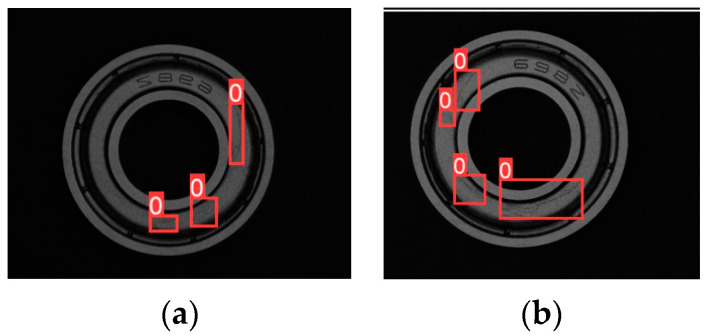
The enhanced image. (**a**) Random scaling and horizontal flipping. (**b**) Random scaling and panning.

**Figure 11 sensors-24-03002-f011:**
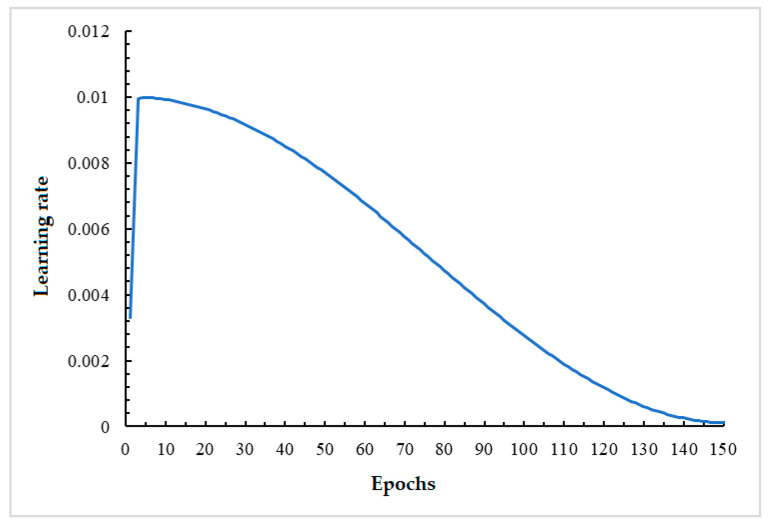
The change in learning rate during training.

**Figure 12 sensors-24-03002-f012:**
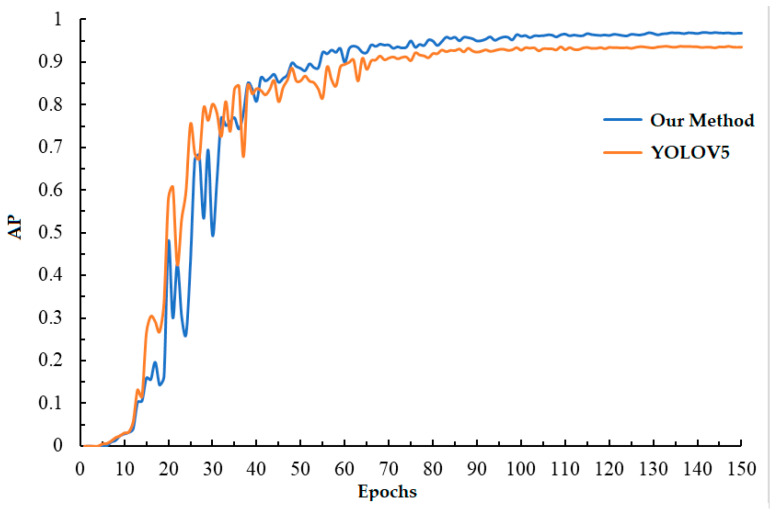
The change in AP in validation set.

**Figure 13 sensors-24-03002-f013:**
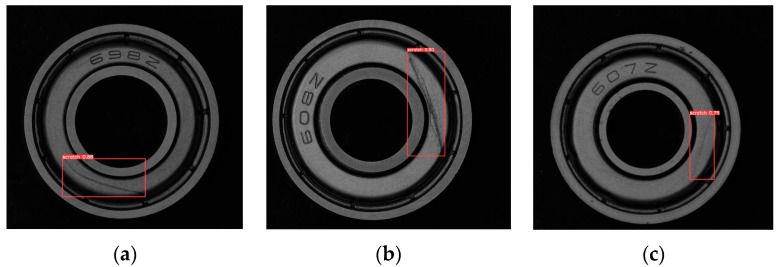
The result of scratches on the bearing surface. (**a**) Test bearing 1. (**b**) Test bearing 2. (**c**) Test bearing 3.

**Table 1 sensors-24-03002-t001:** Parameters of industrial camera used in this study.

Performance	Parameter
model	TC23036
sensor type	CMOS
sensor model	IMX264
pixel size	3.45 µm
target size	2/3″
resolution	2448 × 2048
data interface	USB3.0
Lens interface	C

**Table 2 sensors-24-03002-t002:** Parameters of lens.

Performance	Parameter
model	TC23036
magnification/×	0.243
aperture diameter/mm	11.0
target size	2/3″
field of view/(mm × mm)	34.98 × 29.18
working distance/mm	102.5
typical distortion (max)/%	<0.04
depth of field/mm	7.00
interface	C

**Table 3 sensors-24-03002-t003:** Parameters of coaxial light source.

Performance	Parameter
color	Blue
housing size/mm	80 × 80
luminous surface size/mm	73 × 73
power consumption	24 v/12 W

**Table 4 sensors-24-03002-t004:** Dataset information.

Dataset	Number of Images	Number of Instances
Training	1206	3015
Validation	206	626
Testing	397	1058

**Table 5 sensors-24-03002-t005:** Parameter configuration of the experiment.

Experimental Environment	Versions
System	Ubuntu 18.04.6 LTS
GPU	NVIDIA Tesla T4
CPU	Intel(R) Xeon(R) CPU @ 2.00 GHz
GPU computing framework	CUDA 11.2
Programming language	Python 3.8.16
Deep learning framework	Pytorch 1.13.0

**Table 6 sensors-24-03002-t006:** Detection results of each model.

Model	Precision/%	Recall/%	F1 Score	AP/%
YOLOV5-CDG	97.2	92.5	0.948	97
YOLOV5	95.9	88	0.918	93.6
YOLOv5 + CA	96.8	92.1	0.944	95.8
YOLO + DCN	96.3	91.9	0.94	95.4
YOLO + Ghost	94.8	87.4	0.909	92.8

**Table 7 sensors-24-03002-t007:** Calculation time of each model.

Model	Number of Participants/M	FLOPs/G	CPU Processing Time/ms	GPU Processing Time/ms
YOLOV5-CDG	4.59	6.8	263.4	12.2
YOLOV5	7.01	15.8	371.8	16.8
YOLOV5 + CA	7.04	15.8	392.8	17.1
YOLO + DCN	7.13	12.7	356.9	15.2
YOLO + Ghost	4.45	9.8	260.8	11.5

**Table 8 sensors-24-03002-t008:** Comparison of the accuracy of different methods.

Model	Accuracy
YOLOV5-CDG	97.00%
Zhengyan G [20]	93.33%
NDT [22]	97.50%
Zheng Z [24]	88.6%

## Data Availability

Data are available on request due to restrictions, e.g., privacy or ethical.

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
