# Peer review of "Research on Bearing Surface Scratch Detection Based on Improved YOLOV5"

_sensors, 2024, doi:10.3390/s24103002_

Round 1

Reviewer 1 Report

Comments and Suggestions for Authors

The methodology description in this manuscript is clear, but lacks comparison with the state-of-the-art. Detailed suggestions are as follows:

1. Provide the extensions for CA and DCN the first time they are mentioned.

2. Line 32 is missing punctuation.

3. The internal logic of sentences from lines 38-40 is peculiar.

4. It is recommended that the authors add a paragraph at the end of the introduction describing the logical flow and organization of the manuscript.

5. Ensure consistency in font size between the main text and figures, for example, Figure 5.

6. It is suggested that the authors highlight the modified parts in Figure 7, while the original YoloV5 framework using a uniform dark color.

7. What is the output of YoloV5? From Figure 1, it seems that only different sizes of convolutional features (green cuboids) are output.

8. The authors' experiments only compare with the unimproved and partially improved YoloV5. It is recommended that the authors include at least three state-of-the-art comparisons focusing on bearing scratch detection to demonstrate the superiority of the proposed network in this task.

Comments on the Quality of English Language

Moderate editing of English language required

Reviewer 2 Report

Comments and Suggestions for Authors

The article is devoted to the problem of bearings’ external surface scratch defect detecting during bearings production and manufacturing. A fairly popular approach based on machine learning technologies for video- and photo- data processing is used for this purpose. The authors propose the improved YOLOV5 network model, called by them as YOLOV5-CDG that has the best characteristics in terms of accuracy and speed in comparison with existing analogues. The material in the article is structured quite well and, in my opinion, is clearly stated. At the same time, with overall positive assessment of the work, I do not understand why the authors chose the “Sensors” journal to publish their results. Unfortunately, the article does not contain any descriptions of computer vision systems (e.g. image sensors or vision/camera-based sensors) for which the algorithms are designed. In my opinion, without such technical details it is difficult to classify the article limited only to image processing as a paper that corresponds to the journal’s scope. Perhaps the authors should supplement the manuscript with appropriate hardware description, as well as the features of obtaining of object images and preliminary processing (filtering, etc.).

There are also some small editorial comments:

- I suggest the authors to correct and extend the section «keywords». In current version, in my opinion, it does not reflect the specific of the article ‑ one keyword forms the name of the prototype of neural network model that is improved by the authors, and the other two are almost identical in meaning;

- There are some abbreviations in text without decryption. For example, CA, DCN in abstract, CNN in line 50 (the decryption is given by two lines above (line 48), but the abbreviation is not entered), etc.

As for me, the article requires a revision.

Reviewer 3 Report

Comments and Suggestions for Authors

This manuscript presents an investigation into the detection of scratches on the surface of bearings. To detect bearing surface defects using scratch images, the authors propose an improved YOLOV5 grid (YOLOV5-CDG). In order to achieve the detection of bearing surface scratches, a dataset of bearing surface scratches developed by the authors is used as a basis. The results obtained with the proposed method are sufficient to satisfy the requirements for detecting scratches on bearing surfaces in industrial environments.

The paper is interesting research, with a right methodology and the manuscript is clear, well organized and structured and the authors have worked exhaustively, taking care of the technical details.

The reviewer considers the manuscript to be satisfactory overall, although a few suggestions for improvement have been made:

- The introduction could be improved by incorporating all relevant studies to provide a comprehensive scientific framework.

- It would be advisable to make a list of acronyms.

- It would be advisable not to include acronyms in the abstract.

For these reasons, the reviewer suggests the manuscript for the publication after minor revisions.

Round 2

Reviewer 1 Report

Comments and Suggestions for Authors

The authors needs to proofread their manuscript to ensure the correctness and fluency of English. Apart from this, I have no further comments. 

Reviewer 2 Report

Comments and Suggestions for Authors

The authors have corrected my comments. The article can be published in Sensors.